# Comparison Between Medetomidine and a Medetomidine–Vatinoxan Combination on Cardiorespiratory Variables in Dogs Undergoing Ovariectomy Anesthetized with Butorphanol, Propofol and Sevoflurane or Desflurane

**DOI:** 10.3390/ani14223322

**Published:** 2024-11-19

**Authors:** Francesca Cubeddu, Gerolamo Masala, Francesca Corda, Andrea Corda, Giovanni Mario Careddu

**Affiliations:** Department of Veterinary Medicine, University of Sassari, 07100 Sassari, Italy; gmasala1@uniss.it (G.M.); francescacorda91@tiscali.it (F.C.); acorda1@uniss.it (A.C.)

**Keywords:** anesthesia, dog, sedation, vatinoxan, medetomidine, inhalant agents, heart rate, mean arterial blood pressure, recovery score, monitoring

## Abstract

Medetomidine is a sedative commonly employed in combination with other anesthetics in anesthesia protocols for small animals, but it has the unwanted side effects of markedly increasing blood pressure and decreasing heart rate. Vatinoxan has been shown to counteract such side effects of medetomidine in experimental settings. In this study, the effects of a medetomidine–vatinoxan combination administered to 20 dogs were compared with those of medetomidine alone administered to as many 20 dogs during a short-term, non-experimental surgical procedure. The anesthetic protocol also included a comparison of two other anesthetics, sevoflurane and desflurane. The results show that vatinoxan is also effective in a clinical setting and encourages its inclusion in anesthetic protocols for dogs in which bradycardia and an increase in systemic pressure should be avoided. Further clinical studies are needed to manage the short periods of hypotension, as well as the slight reduction in sedative and pain-relieving medetomidine effects found, particularly when vatinoxan is in combination with desflurane rather than sevoflurane.

## 1. Introduction

Medetomidine is a potent and selective α2-adrenoceptor agonist widely used for sedation and analgesia of animals [1]. Despite the advantageous effects that medetomidine exerts on the central nervous system, activation of peripheral α2-adrenergic receptors causes peripheral vasoconstriction and an increase in systemic vascular resistance (SVR) with a consequent increase in systemic pressure and a decrease in heart rate (HR) [2,3]. This causes a reduction in cardiac output (CO) and oxygen (O_2_) delivery to organs such as the heart, although myocardial O_2_ supply–demand coupling remains unchanged because myocardial O_2_ demand following medetomidine is reduced from the reduction in HR and contractility [4].

Vatinoxan (formerly known as MK-467 and L-659’066) is a peripherally selective α2-adrenoceptor antagonist used in combination with medetomidine to antagonize the peripheral α2-receptoral effects of medetomidine [5,6,7]. Vatinoxan poorly penetrates the blood–brain barrier in mammals [8] and reduces in a dose-dependent manner the peripherally mediated vasoconstriction and bradycardia caused by medetomidine while maintaining the central sedative action in dogs [9,10]. Concomitant administration of vatinoxan and medetomidine resulted in reduced SVR and systemic arterial blood pressure (sAP), increased HR, and improved CO compared with treatments without vatinoxan [11,12,13].

Sevoflurane and desflurane are the newer available inhalant agents for the maintenance of general anesthesia, featuring a rapid pharmacokinetic profile. Both tend to reduce sAP, primarily by decreasing SVR [14,15]. Sevoflurane maintains a relatively stable HR, whereas desflurane increases it by activating the sympathetic nervous system [16].

Butorphanol is a synthetic opioid commonly administered with medetomidine to improve the level and quality of sedation and analgesia in dogs [2,17,18]. Butorphanol induces mild hypoventilation, less than pure µ-agonist opioids [19]. When administered alone, butorphanol causes a small but significant decrease in HR, sAP, cardiac index, and arterial O_2_ partial pressure [20,21], as well as a slight increase in the SVR index [20].

Propofol is a sedative–hypnotic agent that induces depression by enhancing the effects of the inhibitory neurotransmitter gamma-aminobutyric acid [22]. Because of its rapid onset of action and rapid renal elimination, propofol is commonly used as a short-acting induction agent [23].

The aim of this study was to compare the cardiorespiratory effects of an intravenous (IV) medetomidine–vatinoxan combination versus an equal dose of medetomidine in a short-term surgery in dogs anesthetized with butorphanol, propofol, and maintained with sevoflurane or desflurane. The hypothesis was that vatinoxan reduces medetomidine-induced bradycardia and ensures adequate antinociception in a short-term surgery, although lowering sAP, especially in anesthesia maintained with desflurane rather than sevoflurane.

## 2. Materials and Methods

### 2.1. Study Design

A prospective, randomized, blinded clinical trial was carried out in healthy bitches undergoing elective open ovariectomy at the Veterinary Teaching Hospital (VTH) of the University of Sassari from September 2023 to August 2024. The study protocol was approved by the Ethical Committee of the same University (prot. N. 0101145/2023). The dog owners authorized the inclusion of the animals in the study by signing an informed consent form.

### 2.2. Animals

Forty female dogs assigned to the ASA I status according to the classification by the American Society of Anesthesiologists based on anamnesis, clinical examination, complete blood cell count, biochemistry profile, and ECG examination were included in the study. Ten dogs were then randomly assigned to each of the following 4 treatment groups:Mede-Sevo: medetomidine 0.25 mg m^−2^ (Sedator^®^ 1.0 mg/mL, Dechra, Turin, Italy) and sevoflurane (Sevoflurane, Baxter S.p.a., Rome, Italy).Mede-Des: medetomidine 0.25 mg m^−2^ and desflurane (Suprane, Baxter S.p.a., Rome, Italy).Vati-Sevo: medetomidine 0.25 mg m^−2^ + vatinoxan 5 mg m^−2^ (Zenalpha^®^ 0.5 mg/mL + 10 mg/mL, Dechra, Turin, Italy) and sevoflurane.Vati-Des: medetomidine 0.25 mg m^−2^ + vatinoxan 5 mg m^−2^ and desflurane.

The sample size was calculated based on the results of a previous study [12] and from our preliminary observations. A sample size of 10 dogs per group was required to detect a mean [±standard deviation (SD)] difference of 20 (±15) beat min^−1^ in HR and of 20 (±15) mmHg in MAP, with 80% power and a significance level of 0.05. Sample size was calculated using the G*Power software, version 3.1.9.6 (Heinrich-Heine-Universität Düsseldorf, Düsseldorf, Germany).

The dogs were admitted to the VTH the evening before the surgical procedures to allow for all pre-anesthetic evaluations and shaving of the skin of the abdomen and right metatarsal in preparation for surgery and metatarsal arterial access, respectively. The dogs were deprived of food overnight, leaving water ad libitum. The following morning, they were acclimatized before the trial in the preparation room for 60 min.

The body surface area (BSA) was calculated using the following formula: BSA (m^2^) = 10.1 × (body weight in kg)^0.67^ × 10^−2^ [24,25,26]. A medetomidine dose of 0.25 mg m^−2^ is equivalent to 0.012 mg kg^−1^ for a 10 kg dog and to 0.009 mg kg^−1^ for a 25 kg dog. Each morning of the study, a veterinarian not involved in the trial drew a paper note from an envelope containing 10 cards for each treatment and filled the syringe containing the drugs under test.

### 2.3. Anesthesia and Instrumentation

Butorphanol (Dolorex^®^ 10 mg/mL MSD Animal Health, Milan, Italy) 0.2 mg kg^−1^ was IV injected through a right cephalic vein access (20-gauge, 32 mm catheter Delta Med, Mantua, Italy). Anesthesia was induced 5 min later with propofol IV (Proposure^®^ 1%, Merial, S.p.a., Mantua, Italy) 5–8 mg kg^−1^ to allow tracheal intubation. The endotracheal tube was connected to the rebreathing circle system of the workstation (Fabius GS, Dräger^®^ Medical S.p.A., Milan, Italy), which provided measurements of respiratory rate (RR), tidal volume (VT), and respiratory minute volume (VM). Sevoflurane or desflurane in O_2_/air was delivered through an out-of-circuit vaporizer with fresh gas flow set to 1.0 L min^−1^ adjusted to maintain O_2_ inspiratory fraction (FiO_2_) at 0.40. End-tidal percentage of sevoflurane (FE’Sevo) or desflurane (FE’Des) was maintained at 1.8% or 6.1%, respectively, equivalent to 0.8 multiple of minimum alveolar concentration (MAC) in dogs. The value of 2.27% was established as the MAC of sevoflurane resulting from the average between 2.36% [27], 2.39% [28], 2.10% [29], and 2.24% [30], while 7.64% is the MAC of desflurane calculated by Pypendop and Ilkiw (2006) [31]. Oxygen inspiratory fraction, end-tidal carbon dioxide partial pressure (PE’CO_2_), FE’Sevo, and FE’Des were detected by the gas module (Scio four Oxi Plus, Dräger^®^ Medical S.p.A., Milan, Italy) and displayed on a monitor (Infinity Delta, Dräger^®^, Lubeck, Germany). Calibration of the gas module was verified daily (5% Sevoflurane Balance Nitrogen Monitor Mix Grade, Size UD Steel Disposable Cylinder, CGA 165, Airgas Part #:Z02NI95MUDC01 and 2% desflurane, 2.5% nitrogen, 5% carbon dioxide, 36% nitrous oxide, Balance Oxygen Monitor Mix Grade Size SD Steel Disposable Cylinder, CGA 600, Airgas Part #:Z05NI2MSDC000).

Right dorsal pedal artery access was obtained (22-gauge, 25 mm catheter Delta Med, Mantua, Italy), and after a second unsuccessful attempt, the dog was excluded from this study. The artery catheter was connected via non-compliant 0.9% saline-filled tubing to the transducer (TranStar^®^ MX9505, Medex, Smiths Medical, Hranice, Czech Republic) for invasive mean blood arterial pressures (MAP) and HR detection (Monitor Infinity Delta, Dräger^®^, Lubeck, Germany). On the same monitor, the peripheral arterial O_2_ saturation (SpO_2_) detected on the tongue, as well as body temperature (BT) detected by an esophageal probe, was displayed. The dog was positioned in dorsal recumbency onto a resistive heating mat, and the pressure transducer was positioned level to the point of the shoulder and zeroed to atmospheric pressure.

Three minutes after FE’Sevo or FE’Des was 0.8 MAC and FiO_2_ was 0.40, medetomidine or medetomidine–vatinoxan diluted to 5 mL with 0.9% NaCl solution was blindly IV administered, always by the same veterinarian anesthetist over 10 s in dogs of respective group treatments. Fifteen minutes later, while scrubbing and preparation of the surgical field were carried out, FE’Sevo or FE’Des was increased to 1.3 MAC (2.95% and 9.93%, respectively) and surgery began. A median celiotomy of 5–6 cm just caudal to the umbilicus was always performed by the same team for exteriorization of the left and then right uterine horn. The necessary traction force for ligation of the ovarian pedicle and horn apex was exerted, and the subsequent ovariectomy was performed. The abdominal wall was closed with three continuous sutures on fascia, subcutis, and skin, respectively.

In our pilot studies (data not published), more than a 20% rise in cardiorespiratory values was observed at ovarian pedicle traction in most dogs with FE’Des raised to only 1.1 MAC. We therefore decided to increase both FE’Sevo and FE’Des to 1.3 MAC, even if such an increase in cardiorespiratory values seldom occurred in dogs maintained with sevoflurane. Fentanyl (Fentadon^®^ 0.5%, Dechra, Turin, Italy) 0.005 mg kg^−1^ was IV-administered as intraoperative rescue analgesia if the animal exhibited signs of pain to surgical stimulation, defined as more than a 20% increase in two of the variables: RR, HR, and MAP [32]. Fentanyl administration was recorded as yes or no. Norepinephrine (Noradrenalina^®^ 0.1%, S.A.L.F. Bergamo, Italy) 0.1–2.0 µg kg^−1^ min^−1^ was IV-administered if MAP fell below 60 mmHg for >10 min. Norepinephrine administration was recorded as yes or no, and the dog was excluded from this study. The dogs were on spontaneous ventilation with the option of volume-controlled mechanical ventilation after 20 s of apnea.

Ringer lactate solution (S.A.L.^®^ Bergamo, Italy) 5 mL kg^−1^ h^−1^ was IV-administered throughout the entire procedure. Glycemia was measured on the mucosa of the upper lip (OneTouch Select^®^, LifeScan Zurich, Switzerland) 10 min before medetomidine or medetomidine–vatinoxan administration (i.e., short before butorphanol administration) and then at 30, 60, and 120 min after. A two-by-two pooling of the glycemic data from the 4 groups (i.e., Mede-Sevo with Mede-Des versus Vati-Sevo with Vati-Des) was performed to detect any difference in glycemic trends between medetomidine alone and the medetomidine–vatinoxan combination. Glucose at 2.5% in ringer lactate was available for IV administration to dogs with glycemia <70 mg dL^−1^.

The MAP, HR, FE’Sevo, FE’Des, PE’CO_2_, SpO_2_, BT, RR, VT, and VM of each dog were measured at different time points. Tidal volume and VM were indexed to the dog’s body weight (BW) to obtain VT/BW and VM/BW, respectively. The first MAP detection was performed before obtaining arterial access with a non-invasive oscillometric method (NIBP) (PetMAP Graphic II, Ramsey Medical, Tampa, FL, USA).

Data collection time points of variables from T-−1 to T9 are listed in Table 1.

Each variable was compared at each time point between the four groups to observe differences in trends depending on the treatment administered. Furthermore, each variable was compared within the same group along different time points to observe the trend, both over time and following the main painful stimulus of surgery, i.e., the traction of the first (T7) and the second (T8) ovarian pedicles, specifically as follows:-At T−1, RR, HR, and NIBP were detected. Non-invasive MAP value of each dog was subsequently adjusted, multiplying it by the correction factor of 0.88 obtained and dividing the average of the IBP-derived MAP values by the average of the NIBP-derived MAP values simultaneously detected during the surgery.-At T0–T9, the IBP-derived MAP, HR, RR, PE’CO_2,_ VT/BW, VM/BW were measured.

At the end of the surgery, the sevoflurane or desflurane vaporizer was closed, and the dogs breathed 100% O_2_ for 2 min before being disconnected from the breathing circuit and transferred to the recovery room for a 90 min observation. The dogs were extubated after 2 consecutive swallowing movements within a 10 s period.

Atipamezole (Atipam 5 mg mL^−1^, Dechra, Turin, Italy) 0.2 mg kg^−1^ was intramuscularly administered if the dog was deeply sedated and HR < 60 beats min^−1^ persisted after extubation for >10 min. Meloxicam (Metacam^®^ 0.5%, Boehringer, Padua, Italy) 0.2 mg kg^−1^ was IV-administered to dogs of all treatments after the last data collection.

The length of surgery was measured from the skin incision (T6) to the end of the skin suture. The extubation, head lift, sternal recumbency, and standing times were measured starting from the closure of the vaporizer (T9). Both the recovery and post-operative periods began from extubation.

### 2.4. Recovery Quality and Post-Operative Analgesic Management

Recovery score was given at 1 min after extubation with the scoring system (Table 2) described by Hampton 2019 [33], and medetomidine 1 µg kg^−1^ as recovery rescue analgesia was IV-administered in case of struggling or excitement; medetomidine administration was recorded as yes or no and evaluation of the dog continued.

Post-operative pain score was assessed at 15, 30, 60, and 90 min after extubation with the Glasgow Composite Measure Pain Scale-Short Form (CMPS-SF) [34]. Buprenorphine 20 µg kg^−1^ as post-operative rescue analgesia was administered intravenously for a CMPS-SF score of ≥6/24. For dogs that could not yet be walked on a leash, the CMPS-SF had 20 scores, and therefore, rescue analgesia was administered for a score of ≥5/20. Buprenorphine administration was recorded as yes or no, and evaluation of the dog ceased.

All the variables were always detected by the same veterinarian anesthetist who was unaware of the treatment she had administered to the dog.

### 2.5. Statistical Analysis

Qualitative variables were described using absolute and relative frequencies, whereas quantitative variables were summarized as medians and interquartile ranges (IQR). To compare quantitative variables between different groups at multiple time points, the Kruskal–Wallis test was used. Post hoc pairwise comparisons were conducted using the Wilcoxon rank-sum test. For comparisons of quantitative variables measured at multiple time points within the same group, the Friedman test was applied. When significant differences were identified, pairwise comparisons between time points were conducted using the Wilcoxon signed-rank test. All analyses were conducted using SAS/STAT^®^ software (version 9.4, SAS Institute Incorporation, Cary, NC, USA). A significance level of *p* < 0.05 was considered for all analyses.

## 3. Results

No significant differences were found between groups for age, BW, hematocrit (Hct), total serum protein (TP), propofol administered, or surgery length (Table 3).

During this study, SpO_2_ was always above 95%, and BT never dropped below 37.3 °C. Glycemia was always >70 mg dL^−1^, and no need for glucose administration occurred in any dog. Ten minutes before medetomidine or medetomidine–vatinoxan administration, blood glucose concentration in the Mede-Sevo and Mede-Des groups pooled together was not significantly different from that in Vati-Sevo and Vati-Des groups pooled together. Very small blood glucose variations were observed at 30 min, whereas at 60 and 120 min after administration, only the Mede-Sevo and Mede-Des combination showed a significant increase (Table 4).

Values of MAP, HR, RR, PE’CO_2_, VT/BW, VM/BW detected at each time point from T−1 to T9 are listed in Table 5.

Twenty minutes before the start of this study (T−1), no significant difference was evident between the four groups in MAP and HR. At the next detection (T0) after induction, tracheal intubation, and FE’Sevo or FE’Des at 0.8 MAC, a significant decrease in both MAP and HR was evident in each group. At the same time point (T0), both MAP and HR did not differ significantly between the four groups (Figure 1 and Figure 2).

After the administration of medetomidine or medetomidine–vatinoxan, in all groups, MAP began to rise rapidly, while HR fell equally rapidly. In the two medetomidine–vatinoxan groups (Vati-Sevo and Vati-Des), MAP reached its peak at 2 min (T1), while in the two groups with medetomidine alone (Mede-Sevo and Mede-Des), MAP continued to increase and peaked at 3 min (T2). At the same time point (T2), HR showed its lowest values of the entire study in the two groups with medetomidine alone. In subsequent detections at 6 and 15 min (T3 and T4, respectively), when FE’Sevo or FE’Des were still at 0.8 MAC, MAP gradually decreased in all four groups, maintaining significantly different values according to the following order of magnitude: Mede-Sevo > Mede-Des > Vati-Sevo > Vati-Des. The lowest MAP value of the entire study was 46 (44–50) mmHg at T4 in Vati-Des. Hypotension (MAP < 60 mmHg) was also detected in Vati-Sevo and Mede-Des when FE’Sevo or FE’Des had been raised to 1.3 MAC (T5) and at the beginning of surgery (T6). Hypotension never lasted longer than 10 min, and no need for norepinephrine administration occurred in any dog. A significant MAP increase at ovarian pedicle traction (T7 and T8) compared with the previous time point (T6) was detected only in Vati-Sevo and Vati-Des groups. Regarding HR, immediately after administration and until the end of surgery (T1–T9), significantly higher values were detected in the groups with medetomidine–vatinoxan (Vati-Sevo and Vati-Des) compared with those with medetomidine alone (Mede-Sevo and Mede-Des). The highest HR value was 118 (113–122) beats min^−1^ in Vati-Sevo at T4, while the lowest value was 36 (32–40) beats min^−1^ in Mede-Sevo at T2. Already starting from T4 and then up to T9, the HR returned to the normally high pre-administration (T0) values in Vati-Sevo and Vati-Des, significantly higher, although slightly, than those of Mede-Sevo and Mede-Des.

Respiratory rate decreased significantly in all groups after induction (from T−1 to T0), but no significant difference was evident between groups at any time point. In treatments with medetomidine–vatinoxan (Vati-Sevo and Vati-Des), a slight but significant reduction was shown only in the first three time points (T1–T3) after administration of the drugs under test. During surgery, it then fluctuated slightly in all groups between 8 (8–10) and 12 (11–14) breaths per min^−1^. The rare cases of bradypnea were short-lived; no cases of apnea occurred, and no dog required mechanical ventilation. The trend of VT/BW did not show significant variations during the entire trial. VM/BW showed a slight but significant decrease in the Vati-Sevo and Vati-Des groups only at the first three time points (T1–T3) after the administration of the drugs under examination. PE’CO2 ranged between 49 (46–52) and 54 (53–56) without significant variations during the procedure or significant differences between the groups.

Fentanyl as intraoperative rescue analgesia had to be administered at the first ovarian pedicle traction (T7) to three dogs of Vati-Des and to one dog of both Vati-Sevo and Mede-Des treatments.

Recovery quality showed the worst scores in Vati-Des, followed by Mede-Des treatment, with significant differences compared with the others (Figure 3).

As recovery rescue analgesia, medetomidine had to be administered to all dogs of Vati-Des and to two dogs of Mede-Des.

Dogs of Mede-Sevo and Vati-Sevo significantly showed the best post-operative pain score. Five dogs in Mede-Des, other than those who had received rescue analgesia 15 min post-operatively, showed mild and transient struggling and excitement. Mild signs of discomfort persisted at 30 and 60 min in Mede-Des and Vati-Des (Table 6). No need for post-operative rescue analgesia or atipamezole administration occurred in any dog.

Extubation time after closing the vaporizer was significantly shorter in treatments with desflurane than in those with sevoflurane. After recovery, rescue analgesia was administered as above; head lift, sternal recumbency, and standing times showed significantly different values according to the following order of magnitude: Vati-Sevo > Mede-Des > Vati-Des > Mede-Sevo (Table 7).

## 4. Discussion

The present work describes how the administration of a medetomidine–vatinoxan combination changes MAP, HR, RR, PE’CO_2_, recovery, and post-operative quality compared with the administration of medetomidine alone and in dogs anesthetized with butorphanol, propofol, and maintained with sevoflurane or desflurane for a routine short-term surgery such as ovariectomy. Most of the studies on vatinoxan in dogs were performed in laboratory beagles under controlled and experimental conditions. Only very few studies were carried out in clinical situations [26,35].

The same medetomidine IV dose of 0.25 mg kg^−1^ as in Salla’s work (2022) was administered in combination with vatinoxan and was used in our study for all treatments without halving it in those with medetomidine alone [12].

The dose of propofol required for successful intubation fell within the values reported in other studies in which premedication consisted of only butorphanol [12,36]. The inclusion of sevoflurane and desflurane in the present study allowed the comparison of some of the effects of these two newer inhalant agents in an anesthetic protocol using vatinoxan in a clinical setting.

The arterial access obtained before the administration of the drugs under test allowed the observation of MAP changes from the very first moments in a clinical situation, an opportunity that was missing in previous clinical studies on vatinoxan [26,35]. During routine surgeries, the commonly measured cardiovascular variables on which intraoperative hemodynamics are assessed are HR and sAP [37]. Cardiac output is usually assessed only under experimental conditions. Even when blood pressure is well maintained after medetomidine, stroke volume, CO, and possibly perfusion may be severely reduced [38]. Experimental studies in dogs sedated with medetomidine or medetomidine–vatinoxan demonstrated that HR was linearly related to CO [38]. Furthermore, concurrent administration of vatinoxan and medetomidine resulted in significantly lower SVR and MAP and significantly higher HR and CO compared with values after administration of medetomidine alone [9]. From the above, it is therefore presumable that even in the few time points (T4, T5, and T6) when hypotension was detected in our work in some dogs treated with medetomidine–vatinoxan, CO was supported by the contextual high and normal HR.

The MAP increase detected at 2 min after administration (T1) in medetomidine–vatinoxan, even if significantly lower in comparison with medetomidine alone, agrees with the finding that vatinoxan did not prevent the initial hemodynamic effects of medetomidine; however, it attenuated them and shortened their duration [13,36]. In subsequent measurements from 3 to 15 min (T2–T4), the lower MAP found in treatments with desflurane compared with those with sevoflurane (Mede-Des vs. Mede-Sevo and Vati-Des vs. Vati-Sevo) agrees with Ryu’s findings in humans that at equi-anesthetic concentration, desflurane exhibits more potent vasodilatory properties than sevoflurane, resulting in higher perfusion index and lower sAP [15]. The very low MAP detected in Vati-Des at 15 min from administration (T4) could be due to a mutual exacerbation of the SVR-reducing effect of both vatinoxan and desflurane [39]. No clinically significant hypotension was observed in dogs administered vatinoxan in combination with sevoflurane 1.0 MAC [15,40]. The MAP increase detected in the medetomidine–vatinoxan treatments at the beginning of the strong painful stimuli during surgery (T7) was most likely evidence of nociception and could be related to a reduction in the analgesic effect exerted by vatinoxan on medetomidine [38,41]. Furthermore, the greater MAP increase observed in Vati-Sevo compared with Mede-Sevo at the same beginning of strong painful stimuli (T7) agrees with the experimental demonstration in dogs that vatinoxan attenuates the MAC reduction in sevoflurane exerted by dexmedetomidine [40,42]. The similar greater MAP increase observed in Vati-Des compared with Mede-Des could demonstrate that vatinoxan also attenuates the dexmedetomidine-induced MAC decrease in other inhalant agents, such as desflurane. This also agrees with the demonstration that vatinoxan reduced the plasma concentration of medetomidine by increasing its clearance and leading to reduced length of sedation and antinociception [35,43]. When vatinoxan is administered in combination with medetomidine, a higher dose of medetomidine might be necessary to achieve the same sedation when compared with the use of medetomidine alone [38]. Mean arterial pressure, HR, and RR values measured at traction of the second ovarian pedicle (T8) were presumably influenced to some extent by the administration of fentanyl at traction of the first one (T7), although this affected only five dogs in three groups. Higher HR detected in medetomidine–vatinoxan compared with medetomidine-alone treatments is commonly found in numerous studies [5,9,10,12,26,35,40].

Although SpO_2_ was always >95%, the reduction in RR and increase in FE’CO_2_ during this study indicate a moderate state of hypoventilation, which is consistent with general anesthesia in spontaneous ventilation. A blood gas analysis would have revealed the true blood content in CO_2_ and O_2_.

The rise in glycemia occurred in dogs administered with medetomidine and not in those administered with medetomidine–vatinoxan seems to confirm that the concomitant administration of vatinoxan to dogs administered with dexmedetomidine prevented major changes in plasma glucose [44].

Compared with Salla’s work (2022) [26], the shorter extubation time in treatments with sevoflurane (7–10 vs. 16–17 min) could have been facilitated by maintaining the ET above 37 °C with the heating mat, even though FE’Sevo was higher (1.3 vs. 1.0 MAC) and the means of surgery length were very similar (50–52 vs. 46.1–47.3 min) [26,45,46]. The shorter extubation time in treatments with desflurane compared with sevoflurane could be related to the faster washout of desflurane [47,48]. The worse recovery score of the Vati-Des treatment compared with the others could also be related to the faster washout of desflurane, along with a reduction in vatinoxan-induced sedation [13,40,49]. Longer head lift time in treatments with sevoflurane compared with those of Salla’s work (2022) (14–27 vs. 5–6 min) may be due to the higher FE’Sevo administered in our work [26]. Head lift, sternal recumbency, and standing times measured in the Vati-Des and Mede-Des groups were to some extent prolonged by the medetomidine administration after extubation to all dogs of Vati-Des and in two dogs of Mede-Des, even if at the low dose of 0.001 mg kg^−1^.

This work has several limitations. The anesthesiologist who collected the cardiovascular data during this study could have potentially inferred the treatment administered from the HR and MAP values they were detecting. The four groups of dogs were of different breeds, weights, and ages, although the last two were without statistical significance. The lack of data on CO has omitted the knowledge of possible organ dysfunction, even during brief periods of hypotension. The anesthetic protocol for an ovariectomy should have included a strong opioid, which would have reduced nociception and inhalant agent requirement but influenced the evaluations and the results of the drugs under test.

## 5. Conclusions

This study confirms the hypothesis that IV-administered vatinoxan in an anesthetic protocol with butorphanol, propofol, and medetomidine and maintained with sevoflurane or desflurane counteracts the bradycardia induced by medetomidine and restores pre-administration HR values within 5–6 min and throughout surgery. At the same time, vatinoxan significantly reduces MAP, especially in anesthesia maintained with desflurane rather than sevoflurane. These results encourage the use of vatinoxan in the clinical setting, particularly in anesthetic protocols for dogs in which the anesthetic strategy should avoid bradycardia and increases in systemic pressure [50]. Further studies are needed to manage the short period of hypotension and to restore the reduced antinociceptive efficacy and duration of the sedative effect of medetomidine induced by vatinoxan.

## Figures and Tables

**Figure 1 animals-14-03322-f001:**
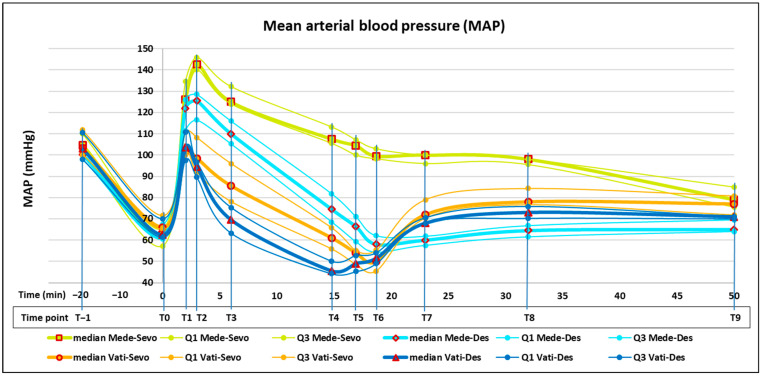
Line graph for median, first quartile (Q1), and third quartile (Q3) of mean arterial blood pressure (MAP) in dogs before anesthesia induction (T−1) and before (T0) and after (T1–T9) administration with medetomidine 0.25 mg m^−2^ and sevoflurane (Mede-Sevo) (n = 10), medetomidine 0.25 mg m^−2^ and desflurane (Mede-Des) (n = 10), medetomidine 0.25 mg m^−2^ + vatinoxan 5 mg m^−2^ and sevoflurane (Vati-Sevo) (n = 10), medetomidine 0.25 mg m^−2^ + vatinoxan 5 mg m^−2^ and desflurane (Vati-Des) (n = 10) during ovariectomy (T6–T9). Time points from T−1 to T9 are plotted along chronological time from 20 min before administration to 50 min after.

**Figure 2 animals-14-03322-f002:**
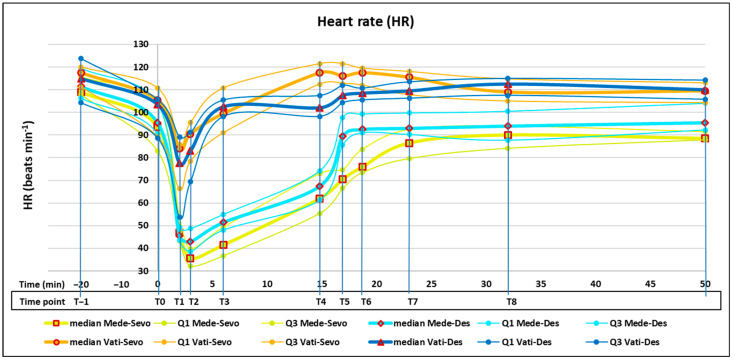
Line graph for median, first quartile (Q1), and third quartile (Q3) of heart rate (HR) in dogs before anesthesia induction (T−1) and before (T0) and after (T1–T9) administration with medetomidine 0.25 mg m^−2^ and sevoflurane (Mede-Sevo) (n = 10), medetomidine 0.25 mg m^−2^ and desflurane (Mede-Des) (n = 10), medetomidine 0.25 mg m^−2^ + vatinoxan 5 mg m^−2^ and sevoflurane (Vati-Sevo) (n = 10), medetomidine 0.25 mg m^−2^ + vatinoxan 5 mg m^−2^ and desflurane (Vati-Des) (n = 10) during ovariectomy (T6–T9). Time points from T−1 to T9 are plotted along chronological time from 20 min before administration to 50 min after.

**Figure 3 animals-14-03322-f003:**
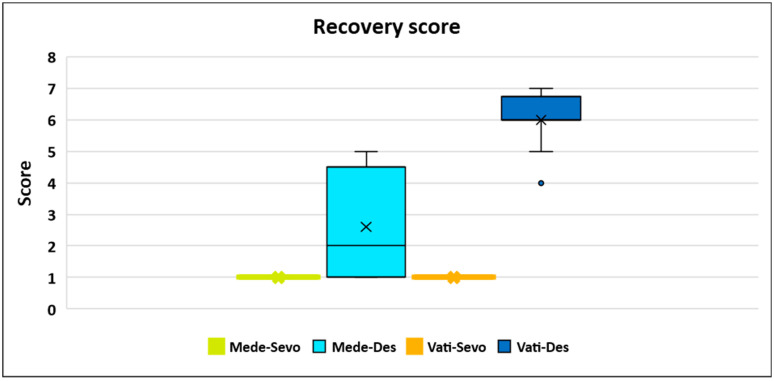
Box and whisker plot for recovery score at one minute after extubation in dogs undergoing ovariectomy administered with medetomidine 0.25 mg m^−2^ and sevoflurane (Mede-Sevo) (n = 10), medetomidine 0.25 mg m^−2^ and desflurane (Mede-Des) (n = 10), medetomidine 0.25 mg m^−2^ + vatinoxan 5 mg m^−2^ and sevoflurane (Vati-Sevo) (n = 10), medetomidine 0.25 mg m^−2^ + vatinoxan 5 mg m^−2^ and desflurane (Vati-Des) (n = 10). The light blue rectangular box (Mede-Des) represents the interquartile range (IQR), spanning from the first quartile (Q1) to the third quartile (Q3), with a line inside indicating the median. In the blue rectangular box (Vati-Des), the median and Q1 overlap. In the green and orange lines (Mede-Sevo and Vati-Sevo, respectively), the median, Q1, and Q3 overlap. The whiskers extend from the box’s edges and denote the non-outlier extremes. Any data points outside this range are considered outliers and are depicted as individual dots.

**Table 1 animals-14-03322-t001:** Data collection time points of variables from twenty minutes before trial (T-−1) to five minutes after closing the vaporizer (T9).

Data Collection Time Points	
T-−1	Twenty minutes before trial
T0	Five minutes after sevoflurane or desflurane 0.8 MAC (1.8% and 6.1%, respectively) and immediately before administration of medetomidine or medetomidine–vatinoxan
T1	Two minutes after administration of medetomidine or medetomidine–vatinoxan
T2	Three minutes after administration of medetomidine or medetomidine–vatinoxan
T3	Six minutes after administration of medetomidine or medetomidine–vatinoxan
T4	Fifteen minutes after administration of medetomidine or medetomidine–vatinoxan
T5	At reaching 1.3 MAC (2.95% sevoflurane or 9.93% desflurane)
T6	At skin incision
T7	At first ovarian pedicle traction (left)
T8	At second ovarian pedicle traction (right)
T9	Five minutes after end of surgery and closing the vaporizer

**Table 2 animals-14-03322-t002:** Recovery scoring system adapted from Hampton 2019 [33].

Factors	Assessments	Score
Struggling/ Excitement	none	0
	transient, easily calmed by the investigator’s voice	1
	prolonged (>1 min)	2
	persistent (or requiring restraint)	3
Paddling/ Flailing	none	0
	transient, easily calmed by the investigator’s voice	1
	prolonged (>1 min)	2
	persistent (or requiring restraint)	3
Vocalization	none	0
	transient, easily calmed by the investigator’s voice	1
	prolonged (>1 min)	2
	persistent (or requiring restraint)	3
Administration of rescue drugs	not given	0
	given	3

**Table 3 animals-14-03322-t003:** Median (interquartile range) of age, body weight (BW), hematocrit (Hct), total serum protein (TP), propofol administered, and surgery length in dogs of the 4 treatment groups (Mede-Sevo, Mede-Des, Vati-Sevo, and Vati-Des).

	Treatment Groups				
Variable	Mede-Sevo (n = 10)	Mede-Des (n = 10)	Vati-Sevo (n = 10)	Vati-Des (n = 10)	*p*-Value
Age (month)	12 (12–22)	12 (9–21)	12 (11–20)	12 (12–18)	0.63
BW (kg)	15 (12–18)	15 (12–19)	14 (13–17)	17 (16–18)	0.79
Hct (%)	54 (52–56)	51 (49–55)	53 (50–53)	51 (49–56)	0.79
TP (g dL^−1^)	6.7 (6.2–6.8)	6.7 (5.9–6.9)	6.3 (5.6–7.1)	6.2 (6.0–6.9)	0.70
Propofol administered (mg kg^−1^)	6.6 (6.0–7.0)	6.8 (6.0–7.3)	6.5 (5.7–7.1)	6.1 (5.4–6.8)	0.71
Surgery length (min)	52 (47–54)	50 (44–58)	50 (43–54)	51 (47–56)	0.93

**Table 4 animals-14-03322-t004:** Median (interquartile range) of glycemia before medetomidine 0.25 mg m^−2^ or medetomidine 0.25 mg m^−2^ + vatinoxan 5 mg m^−2^ administration (−10 min) and at 30, 60, and 120 min after (30 min, 60 min, and 120 min, respectively). Dogs of the two groups administered medetomidine alone (Mede-Sevo and Mede-Des) were pooled together (n = 20), as were dogs of the two groups administered medetomidine–vatinoxan (Vati-Sevo and Vati-Des) (n = 20).

	Treatment	Time (min)			
		−10 min	30 min	60 min	120 min
Glycemia (mg dL^−1^)	Mede-Sevo and Mede-Des pooled	89 (84–94)	91 (82–96)	105 (98–114) *^,#^	109 (99–116) *^,#^
	Vati-Sevo and Vati-Des pooled	91 (87–92)	86 (82–94)	85 (80–93)	88 (83–93)

* At a given time point, the value is significantly (*p* < 0.01) higher than that of the other treatment. ^#^ For a given treatment, the value is significantly (*p* < 0.01) higher than that at 10 min (−10 min) before administration of medetomidine or medetomidine–vatinoxan.

**Table 5 animals-14-03322-t005:** Median (interquartile range) of mean arterial pressure (MAP), heart rate (HR), respiratory rate (RR), end-tidal carbon dioxide (PE’CO_2_), tidal volume/body weight (VT/BW), and respiratory minute volume/body weight (VM/BW) from 20 min before trial (T−1) to the end of the ovariectomy (T9) in dogs administered with medetomidine 0.25 mg m^−2^ and sevoflurane (Mede-Sevo), medetomidine 0.25 mg m^−2^ and desflurane (Mede-Des), medetomidine 0.25 mg m^−2^ + vatinoxan 5 mg m^−2^ and sevoflurane (Vati-Sevo), medetomidine 0.25 mg m^−2^ + vatinoxan 5 mg m^−2^ and desflurane (Vati-Des).

Time Point	Treatment	MAP (mmHg)	HR (Beat min^−1^)	RR (Breath min^−1^)	PE’CO_2_ (mmHg)	VT/BW (mL kg^−1^)	VM/BW (mL min^−1^ kg^−1^)
T−1	Mede-Sevo	105 (101–110)	109 (108–116)	22 (19–22)	na	na	na
	Mede-Des	100 (98–103)	112 (106–120)	21 (19–23)	na	na	na
	Vati-Sevo	103 (100–112)	118 (111–120)	23 (20–24)	na	na	na
	Vati-Des	103 (98–111)	115 (104–124)	22 (21–23)	na	na	na
	*p*-value	0.72	0.63	0.70			
T0	Mede-Sevo	63 (57–66) ^a^	94 (83–104) ^a^	10 (8–12)	52 (48–52)	12 (10–14)	105 (79–157)
	Mede-Des	62 (61–67) ^a^	96 (91–104) ^a^	9 (9–12)	50 (49–52)	12 (10–13)	107 (96–156)
	Vati-Sevo	66 (65–72) ^a^	106 (89–111) ^a^	10 (9–13)	50 (49–52)	11 (10–13)	120 (93–133)
	Vati-Des	63 (61–70) ^a^	104 (89–106) ^a^	10 (9–12)	49 (46–52)	12 (10–12)	114 (91–132)
	*p*-value	0.35	0.47	0.93	0.79	0.88	0.96
T1	Mede-Sevo	126 (125–135) ^b^	47 (43–54) ^b^	8 (8–10)	53 (51–54)	11 (9–13)	92 (69–124)
	Mede-Des	122 (111–126) ^b^	46 (44–49) ^b^	8 (8–11)	52 (51–54)	12 (11–12)	91 (85–132)
	Vati-Sevo	102 (100–111) ^b^	84 (66–86) ^b^	7 (6–10) ^b^	53 (51–54) ^b^	10 (10–12) ^b^	89 (54–97) ^b^
	Vati-Des	104 (98–111) ^b^	78 (54–89) ^b^	8 (7–10) ^b^	52 (49–53) ^b^	11 (10–12) ^b^	93 (73–103) ^b^
	*p*-value	0.00 ^2,3,4,5^	0.00 ^2,3,4,5^	0.48	0.52	0.75	0.24
T2	Mede-Sevo	143 (140–146) ^b^	36 (32–40) ^b^	9 (7–11)	53 (52–54)	12 (9–12)	79 (69–123)
	Mede-Des	126 (117–129) ^b^	43 (39–49) ^b^	8 (6–11)	53 (51–55)	11 (11–12)	81 (66–123)
	Vati-Sevo	99 (92–108) ^b^	91 (79–96) ^b^	8 (6–9) ^b^	52 (51–54) ^b^	11 (10–12) ^b^	78 (62–101) ^b^
	Vati-Des	95 (90–97) ^b^	83 (70–91) ^b^	7 (6–10) ^b^	53 (52–55) ^b^	11 (10–12) ^b^	81 (73–103) ^b^
	*p*-value	0.00 ^1,2,3,4,5^	0.00 ^1,2,3,4,5^	0.08	0.68	0.52	0.33
T3	Mede-Sevo	125 (124–132) ^b^	42 (37–49) ^b^	9 (7–10)	54 (52–55)	12 (9–13)	87 (66–128)
	Mede-Des	110 (105–116) ^b^	52 (48–55) ^b^	8 (7–11)	53 (51–55)	11 (11–12)	85 (78–118)
	Vati-Sevo	86 (78–96) ^b^	100 (91–111)	8 (6–9) ^b^	54 (53–54) ^b^	10 (9–12) ^b^	74 (66–85) ^b^
	Vati-Des	70 (63–75)	103 (98–106)	7 (7–9) ^b^	53 (51–55) ^b^	11 (10–12) ^b^	86 (73–97) ^b^
	*p*-value	0.00 ^1,2,3,4,5,6^	0.00 ^2,3,4,5^	0.12	0.74	0.85	0.48
T4	Mede-Sevo	108 (106–113) ^b^	62 (56–73) ^b^	10 (8–11)	53 (52–55)	12 (10–14)	104 (70–148)
	Mede-Des	75 (69–82) ^b^	68 (61–74) ^b^	9 (8–11)	53 (51–55)	11 (10–12)	86 (70–148)
	Vati-Sevo	61 (56–66) ^b^	118 (113–122) ^b^	9 (8–11)	51 (50–53)	11 (10–12)	108 (86–118)
	Vati-Des	46 (44–50) ^b^	102 (98–108)	9 (8–11)	53 (50–55)	11 (9–12)	105 (94–111)
	*p*-value	0.00 ^1,2,3,4,5,6^	0.000 ^2,3,4,5,6^	0.92	0.63	0.73	0.97
T5	Mede-Sevo	105 (100–107) ^b^	71 (67–75) ^b^	9 (8–10)	54 (52–56)	11 (9–13)	104 (71–132)
	Mede-Des	67 (59–71)	90 (86–98) ^b^	8 (8–10)	54 (51–56)	11 (9–12)	87 (73–112)
	Vati-Sevo	54 (49–55) ^b^	116 (113–122) ^b^	8 (8–11)	52 (50–54)	10 (10–12)	93 (74–104)
	Vati-Des	49 (45–53) ^b^	108 (104–112) ^b^	10 (9–12)	54 (52–56)	11 (10–12)	115 (103–121)
	*p*-value	0.00 ^1,2,3,4,5^	0.00 ^1,2,3,4,5,6^	0.24	0.38	0.75	0.25
T6	Mede-Sevo	100 (98–103)	76 (74–84)	9 (8–10)	54 (52–55)	10 (9–12)	88 (64–127)
	Mede-Des	58 (54–62)	93 (91–99)	9 (8–9)	54 (51–55)	11 (10–12)	90 (77–112)
	Vati-Sevo	50 (46–55)	118 (112–120)	9 (8–11)	52 (51–53)	11 (10–12)	109 (92–115)
	Vati-Des	52 (49–54)	109 (106–111)	10 (9–13)	54 (53–56)	12 (10–13)	120 (106–133)
	*p*-value	0.00 ^1,2,3,5^	0.00 ^1,2,3,4,5,6^	0.25	0.24	0.62	0.23
T7	Mede-Sevo	100 (96–100)	87 (80–93) ^c^	10 (8–10)	53 (52–55)	11 (9–12)	90 (70–139)
	Mede-Des	60 (58–62)	93 (90–100)	9 (8–10)	53 (52–55)	11 (10–13)	95 (8–116)
	Vati-Sevo	72 (70–79) ^c^	116 (108–118)	10 (9–11)	51 (49–52)	11 (11–13)	118 (105–128)
	Vati-Des	68 (68–71) ^c^	110 (106–114)	11 (9–13)	54 (52–55)	12 (11–13)	131 (105–150)
	*p*-value	0.00 ^1,2,3,4,5,6^	0.00 ^1,2,3,4,5^	0.19	0.07	0.64	0.14
T8	Mede-Sevo	98 (96–98)	90 (84–94) ^c^	9 (7–10)	54 (52–54)	11 (10–12)	91 (75–114)
	Mede-Des	65 (62–67)	94 (88–101)	10 (9–11)	52 (51–55)	12 (10–13)	114 (93–117)
	Vati-Sevo	78 (77–84) ^c^	109 (105–115) ^c^	10 (9–12)	50 (50–52)	11 (10–13)	124 (102–133)
	Vati-Des	73 (70–76) ^c^	113 (108–115)	11 (9–13)	54 (53–56)	12 (11–13)	129 (114–143)
	*p*-value	0.00 ^1,2,3,4,5,6^	0.00 ^2,3,4,5^	0.19	0.06	0.42	0.10
T9	Mede-Sevo	79 (76–85)	89 (88–92)	9 (9–11)	52 (49–54)	11 (9–13)	108 (78–133)
	Mede-Des	65 (64–70)	96 (92–104)	10 (9–12)	51 (50–52)	13 (10–14)	115 (95–150)
	Vati-Sevo	77 (72–81)	110 (104–113)	11 (10–12)	50 (49–51)	12 (11–13)	136 (118–146)
	Vati-Des	71 (70–71)	110 (106–114)	12 (11–14)	53 (52–54)	13 (12–15)	157 (134–178)
	*p*-value	0.00 ^1,3,4,6^	0.00 ^2,3,4,5^	0.06	0.08	0.18	0.06

^1^ At a given time point, the value of Mede-Sevo is significantly different from that of Mede-Des. ^2^ At a given time point, the value of Mede-Sevo is significantly different from that of Vati-Sevo. ^3^ At a given time point, the value of Mede-Sevo is significantly different from that of Vati-Des. ^4^ At a given time point, the value of Mede-Des is significantly different from that of Vati-Sevo. ^5^ At a given time point, the value of Mede-Des is significantly different from that of Vat-Des. ^6^ At a given time point, the value of Vati-Sevo is significantly different from that of Vati-Des. ^a^ For a given treatment, the value at T0 is significantly different from that at T−1. ^b^ For a given treatment, the value at T1, T2, T3, T4, and T5 is significantly different from that at T0. ^c^ For a given treatment, the value at T7 and T8 is significantly different from that at T6. na, not assessed because the dog was not yet intubated.

**Table 6 animals-14-03322-t006:** Median (interquartile) scores of recovery at 1 min and of post-operative pain at 15, 30, 60, and 120 min after awakening in dogs undergoing ovariectomy administered with medetomidine 0.25 mg m^−2^ and sevoflurane (Mede-Sevo) (n = 10), medetomidine 0.25 mg m^−2^ and desflurane (Mede-Des) (n = 10), medetomidine 0.25 mg m^−2^ + vatinoxan 5 mg m^−2^ and sevoflurane (Vati-Sevo) (n = 10), medetomidine 0.25 mg m^−2^ + vatinoxan 5 mg m^−2^ and desflurane (Vati-Des) (n = 10).

		Treatment Groups				
	Variable	Mede-Sevo (n = 10)	Mede-Des (n = 10)	Vati-Sevo (n = 10)	Vati-Des (n = 10)	*p*-Value
Post-operative pain score	15 min	1 (1-1)	2 (1-2)	1 (1-1)	1 (1-1)	<0.05
	30 min	0 (0-0)	1 (1-1)	0 (0-0)	1 (1-1)	<0.05
	60 min	0 (0-0)	0 (0-0)	0 (0-0)	1 (1-1)	<0.05
	90 min	0 (0-0)	0 (0-0)	0 (0-0)	0 (0-0)	1

**Table 7 animals-14-03322-t007:** Median (interquartile) extubation, head lift, sternal recumbency, and standing times after anesthesia in dogs undergoing ovariectomy administered with medetomidine 0.25 mg m^−2^ and sevoflurane (Mede-Sevo) (n = 10), medetomidine 0.25 mg m^−2^ and desflurane (Mede-Des) (n = 10), medetomidine 0.25 mg m^−2^ + vatinoxan 5 mg m^−2^ and sevoflurane (Vati-Sevo) (n = 10), medetomidine 0.25 mg m^−2^ + vatinoxan 5 mg m^−2^ and desflurane (Vati-Des) (n = 10).

	Treatment Groups				
Variable	Mede-Sevo (n = 10)	Mede-Des (n = 10)	Vati-Sevo (n = 10)	Vati-Des (n = 10)	*p*-Value
Extubation time (min)	10 (9.3–11)	3.5 (3.0–4.0)	7.0 (6.0–8.0)	3.0 (2.3–3.0)	<0.01
Head lift time (min)	27 (26–27)	15 (14–17)	14 (10–14)	16 (14–19)	<0.01
Sternal recumbency time (min)	34 (33–35)	18 (15–18)	16 (13–17)	22 (21–25)	<0.01
Standing time (min)	41 (33–42)	22 (18–25)	20 (19–23)	29 (28–31)	<0.01

## Data Availability

The original contributions presented in the study are included in the article; further inquiries can be directed to the corresponding authors.

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
