# Peer review of "Comparison Between Medetomidine and a Medetomidine–Vatinoxan Combination on Cardiorespiratory Variables in Dogs Undergoing Ovariectomy Anesthetized with Butorphanol, Propofol and Sevoflurane or Desflurane"

_animals, 2024, doi:10.3390/ani14223322_

Round 1
Reviewer 1 Report
Comments and Suggestions for Authors
The work is interesting but some points need to be clarified and the writing improved.
I read with interest the article entitled “ Comparison between medetomidine and a medetomidine-vatinoxan combination on cardiorespiratory variables in dogs undergoing ovariectomy anesthetized with sevoflurane or desflurane.” Some clarifications are necessary before I can consider it for publication.
A limitation of the study could be the presence of a single observer who was not aware of the protocols used, as this could influence the results; a strength of the study is certainly the possibility of learning about the different possibilities in the use of the vatinoxan/medetomidine combination; the methodology used is correct, the materials and methods, results and the interpretation of the data need to be clearer.
Throughout the study, you refer only to the administration of Vatinoxan, and only in a few places do you mention that the product you used is already on the market in combination with medetomidine. It would be appropriate to refer to an 'association' at all times to give the reader greater clarity.
Line 13: The sentence should be clarified: The combination of Vatinoxan/medetomidine was used in comparison with the medetomidine alone.
Line 21: Here again, there will be a need for clarification.
Line 94-102: It is necessary to better describe the timing of drug administration.
Line 106: Who would administer the drug?
Line 111: What was the dose of propofol? How did you get it?
Line 149: Define HR . You can add this reference to enrich the manuscript. DOI: 10.3390/ani12223134.
Line 172: Did the surgery time (minutes) vary between subjects?
Line 346-353: The whole period is unclear. How do you distinguish between the postoperative period and the recovery period?
Line 374: This statement is not entirely correct; let it be clear from the outset that this is a comparison with a drug association.
Author Response
The Responses to Reviewer 1 Comments have been uploaded, containing point by point the details of the revisions to the manuscript.

Reviewer 2 Report
Comments and Suggestions for Authors
The use of vatinoxan, a new peripheral alpha-2 adrenergic antagonist, is very beneficial not only for the professional public, but also for the academic community. The presented paper is very detailed. From this point of view, I consider the article beneficial and up-to-date.
Title
L1-5 – Propofol and butorphanol were also used for anesthesia. I would recommend mentioning both drugs in the title.
Abstract
The abstract does not have the usual structure for scientific articles, I recommend rewriting it.
Keywords
The same words from the title are repeated.
Introduction
L50-51 – I think it is not necessary to mention the trade name at this place, I recommend moving it to Materials and Methods
If you used both Sevo and Des, I would supplement the hypothesis of their influence in the present study.
Materials and methods
2.2. Animals – Why did you put exactly 10 animals in each group? Please complete the Sample Size Collection.
L150 – You did not use norepinephrine on any dog. Is it necessary to mention it?
L153 – You did not use ventilatory support. Is it necessary to mention it?
L163 – How did you measure HR, RR, MAP, PE´CO2, VT and MV? Please describe more precisely.
L195 – How did you measure length of surgery, extubation, head lift, sternal recumbency and standing times? Please describe more precisely.
L208 – "... was IV administered for score 208 ≥5/20 or ≥6/24 ..." – please clarify
Figure 1 – Glasgow CMPS-SF is well known. I don't think it's necessary to give the figure, just a citation is enough.
Results
L227-228 – I think the first two sentences of this paragraph are not necessary, I recommend removing.
L238, 239, 243 – Repeat values ​​from the table, I recommend removing.
Table 4 – Why were blood glucose data pooled? I recommend providing separate data for individual groups.
Table 5 – Why are data at T1, T2 and T3 for RR, PE´CO2, VT/BW and VM/BW missing?
Figures 2 and 3 repeat the data from table 5, I recommend removing the figures.
L295-329 – Repeat values ​​from the table, I recommend removing.
L309-310 – "Hypotension never lasted longer than 10 minutes and no 309 need for norepinephrine administration occurred in any dog.“ – I recommend removing this sentence.
L300-331 – "Fentanyl as intraoperative rescue analgesia had to be administered to 3 dogs of Vati-Des and to 1 dog of both Vati-Sevo and Mede-Des treatments." – How will this affect the results?
L345-350 – "As recovery rescue analgesia, medetomidine had to be administered to all dogs of Vati-Des and in 2 dogs of Mede-Des." – The injection of medetomidine will affect the results mentioned in Table 7 and the statement at L362-363.
Conclusions
In the first sentence, I recommend mentioning the currently used drugs - butorphanol, propofol, Sevo and Des.
L457-460 – “These results … vasoconstriction [50, 51]” – I recommend removing this sentence.
The manuscript is written precisely and brings new interesting facts. I recommend accepting the manuscript after revisions and additions to the text.
Author Response
The Responses to Reviewer 2 Comments have been uploaded, containing point by point the details of the revisions to the manuscript.

Reviewer 3 Report
Comments and Suggestions for Authors
this was a well-done and well-written paper.
I am curious as to why the medetomidine-vatinoxan was administered IV versus IM as part of the premed.
In the USA, the label for dogs is for IM use only.
It is stated that the vatinoxan may decrease the analgesic effect of the medetomidine, do you think it has a shortening effect on the sedation properties?
Author Response
The Responses to Reviewer 3 Comments have been uploaded, containing point by point the details of the revisions to the manuscript.

Round 2
Reviewer 1 Report
Comments and Suggestions for Authors
The work has been improved and is now ready for publication